# Extended X-ray absorption fine structure of dynamically-compressed copper up to 1 terapascal

H. Sio [1] ✉, A. Krygier [1], D. G. Braun[1], R. E. Rudd [1], S. A. Bonev[1], F. Coppari [1], M. Millot [1], D. E. Fratanduono[1], N. Bhandarkar [1], M. Bitter[2], D. K. Bradley[1], P. C. Efthimion[2], J. H. Eggert [1], L. Gao [2], K. W. Hill[2], R. Hood[1], W. Hsing[1], N. Izumi [1], G. Kemp[1], B. Kozioziemski[1], O. L. Landen [1], K. Le Galloudec[1], T. E. Lockard[1], A. Mackinnon[1], J. M. McNaney [1], N. Ose[1], H.-S. Park [1], B. A. Remington[1], M. B. Schneider[1], S. Stoupin[1], D. B. Thorn[1], S. Vonhof[3], C. J. Wu[1] & Y. Ping [1]

Large laser facilities have recently enabled material characterization at the pressures of Earth and Super-Earth cores. However, the temperature of the compressed materials has been largely unknown, or solely relied on models and simulations, due to lack of diagnostics under these challenging conditions. Here, we report on temperature, density, pressure, and local structure of copper determined from extended x-ray absorption fine structure and velocimetry up to 1 Terapascal. These results nearly double the highest pressure at which extended x-ray absorption fine structure has been reported in any material. In this work, the copper temperature is unexpectedly found to be much higher than predicted when adjacent to diamond layer(s), demonstrating the important influence of the sample environment on the thermal state of materials; this effect may introduce additional temperature uncertainties in some previous experiments using diamond and provides new guidance for future experimental design.

Dynamic compression experiments in planar geometry at high-energy-density facilities[1–3] have recently achieved record pressures[4–6] and expanded the frontier for studying material responses under extreme conditions. In particular, shockless or ramp compression enables reaching extraordinary pressure while maintaining the solid state by compressing the sample over a much longer timescale in contrast to shock compression[7,8]. This makes it possible to compare with theoretical predictions of material response and crystalline structure under a wide range of conditions, including those relevant to planetary science and inertial confinement fusion. While much progress has been made in diagnosing these extreme pressure-density states[4,5,9–13], there is a lack of reliable temperature measurements needed to constrain the final thermodynamic state of the compressed material across the full scope of dynamic compression platforms. Temperature measurements by optical pyrometry are limited to transparent materials or apparent surface temperature of opaque materials, and also not applicable below a certain temperature threshold due to detection limit[14,15].

One promising approach for determining the bulk temperature of dynamically compressed solids is extended x-ray absorption fine structure (EXAFS). EXAFS refers to modulations in the X-ray absorption spectrum caused by photoelectron waves scattering off nearby atoms and is a well-established technique to probe local structure as well as structural and thermal disorder[16]. The period of the modulations is set by the interatomic distances, and the modulation amplitude attenuation at higher X-ray energies is related to the Debye–Waller

[1]Lawrence Livermore National Laboratory, 7000 East Ave, Livermore, CA 94550, USA. [2]Princeton Plasma Physics Laboratory, Princeton University, 100 Stellarator Rd, Princeton, NJ 08540, USA. [3]General Atomics, 3550 General Atomics Court, San Diego, CA 92121, USA. ✉e-mail: sio1@llnl.gov

factor[17], which depends on the ion temperature. The EXAFS sensitivity to temperature and phase transformation in nanosecond-scale shock experiments was first demonstrated in V and Ti[18,19] up to 50 GPa, and later in Fe[20] up to 560 GPa[21–23] and Pt up to 325 GPa[24].

In this work, we present the results of experiments investigating the temperature, density, pressure, and structure of copper at the National Ignition Facility (NIF)[3] up to 1 TPa. These EXAFS measurements of dynamically compressed Cu also nearly double the highest pressure at which EXAFS has been reported in any material[21]. It is observed that, across the ramp and shock-ramp loading paths investigated here, the Cu temperature is significantly higher than predictions by our hydrodynamic simulations when the Cu sample is adjacent to diamond layer(s). This finding suggests that the assumption of minimal thermal transport over a several-ns time scale should be examined on a case-by-case basis, and may lead to undesirably large uncertainties when determining phase diagram as well as pressure-density-temperature relationship. Our EXAFS data also shows that Cu remains in the face-centered-cubic (fcc) structure up to 5600 K near 400 GPa, as compared with theoretical calculations predicting a transformation to body-centered-cubic (bcc) structure near this temperature[25,26]. In addition, temperature, density, pressure, and phase are simultaneously constrained using EXAFS and velocimetry, definitively constraining the thermodynamic state on the phase diagram in an equation-of-state experiment. This is in contrast to many previous dynamic-compression experiments where pressure, density, and/or phase are measured, but the temperature is only estimated or modeled using simulations.

## Results

Our experiments are performed at the NIF in the experimental configuration shown in Fig. 1. Sixteen NIF lasers are used to drive the target package with a laser pulse shape to compress the Cu sample along a quasi-isentropic path. Several target designs are used in these experiments, which follow the general pattern of a beryllium ablator, an X-ray heat shield, and then the Cu sample between a pusher in front, and a window at the rear side. Reverberating compression waves in the Cu sample hold the pressure during the X-ray source[27] emission time for the EXAFS measurement. A velocity interferometer system for any reflector (VISAR)[28] system simultaneously measures the velocity either at the Cu-window interface and/or at the window free surface to determine the pressure using the characteristic analysis method[10,29].

### Experiments varying shock strength

Up to 88 lasers (351 nm, 4.8 kJ per beam) are used to drive a Ti foil X-ray source[27] on both sides, producing a bright X-ray continuum around the

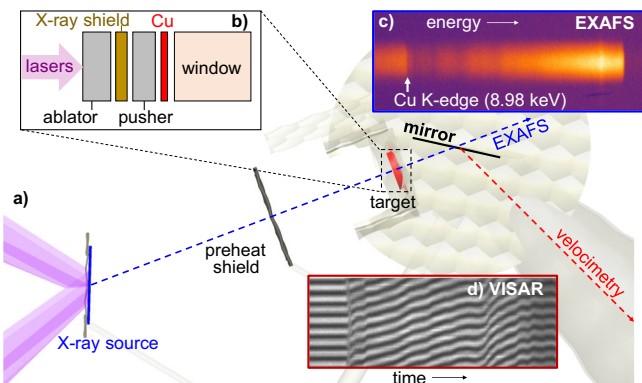

**Fig. 1 | EXAFS experimental setup at the National Ignition Facility. a** The Cu sample in the **b** target package is driven by 16 beams. Up to 88 beams are incident on the Ti X-ray source. **c** X-ray transmission through the target package is measured by the HiRAXS X-ray spectrometer. **d** A mirror behind the target package enables simultaneous velocity measurements of the Cu/window interface and/or window free surface using VISAR.

Cu K-edge energy region. Figure 2a and b show EXAFS data collected on experiments using a target package design with the Cu sample between Al and LiF, and the corresponding laser pulse shapes are plotted in Fig. 2c. In the first experiment (N201222), the Cu was initially shocked to 40 GPa, and then compressed to ~400 GPa. Figure 2d shows the Cu pressure history for this shot derived from analysis of VISAR data. To reach similar pressure but a higher temperature in the second experiment (N200917), a higher initial shock (~85 GPa) is launched into the Cu sample, which is then compressed from a higher initial entropy state to a higher final temperature. The higher-temperature state in the second experiment (N200917) is reflected in the lower peak amplitudes as compared to the first experiment (N201222). The shaded regions in Fig. 2 represent best-fit synthetic EXAFS signals to the measured EXAFS using MD simulations. Temperature uncertainty includes measurement uncertainty (photon statistics), diagnostic calibrations (crystal dispersion relation and flatfield), and the EXAFS amplitude sensitivity to temperature. The best-fit temperatures of these two EXAFS spectra are 2300 ± 300 and 3200 ± 400 K, respectively.

These two experiments clearly demonstrate that different thermal states can be reached using a shaped laser drive in dynamically compressed Cu and that the EXAFS measurements are able to distinguish between the different thermal states. The initial shock launched into the Cu sample is 40 GPa and 85 GPa, respectively, for N201222 and N200917, corresponding to an initial shock heating difference of ~800 K (using the Cu equation of state SESAME 3336). This is in very good agreement with the EXAFS temperature difference between these two experiments.

### Adjacent layer effect on temperature

Next, we compare three experiments with different material layers adjacent to the Cu sample at similar pressures and densities (near 400 GPa at ~16.0 g cm⁻³). At a given density, higher EXAFS amplitudes generally correspond to lower temperatures. The experimental data in Fig. 3a–c are presented in the order of increasing temperature. The Cu sample pressure histories for these three experiments are shown in Fig. 3d.

The first experiment in Fig. 3a is the same as Fig. 2a and shown here as a comparison. This experiment with a LiF window and a ramped drive presents the lowest temperature among the three EXAFS spectra in Fig. 3. The following two EXAFS experiments that both used a diamond window in Fig. 3b and c also use a ramp drive. In N200520, the Cu sample is between Be and diamond, and the EXAFS amplitudes are noticeably lower than those of a similarly driven ramp shot using LiF (N201222). In the last experiment (N210719), the Cu sample is between two diamond layers and ramp-compressed to a similar pressure. This EXAFS measurement of the Cu sample between two diamond layers produces the lowest EXAFS amplitudes, indicating the highest Cu temperature.

The best-fit temperatures of these three EXAFS spectra are 2300 ± 300, 3900 ± 400, and 5600 ± 500 K, respectively for N201222, N200520, and N210719. The measured pressure time histories for the EXAFS data in Fig. 3a and c both show a similar initial shock at ~40 GPa. Therefore, this temperature difference cannot be explained by the initial shock strength. Possible mechanisms for the abnormal heating with adjacent diamond layers are discussed in a later section.

### Copper lattice structure

We also examine the EXAFS spectra in these NIF experiments to determine the Cu crystal structure at pressures near 400 GPa and 1 TPa. The EXAFS oscillatory patterns are sensitive to the local atomic arrangement and thus can distinguish between crystal structures with significantly different first-shell configurations, such as between fcc and bcc lattices of Cu. Theoretical calculations[26] and quantum molecular dynamics simulations[25] predicted that the bcc phase is stabilized at a high temperature near 400 GPa where we performed the highest-

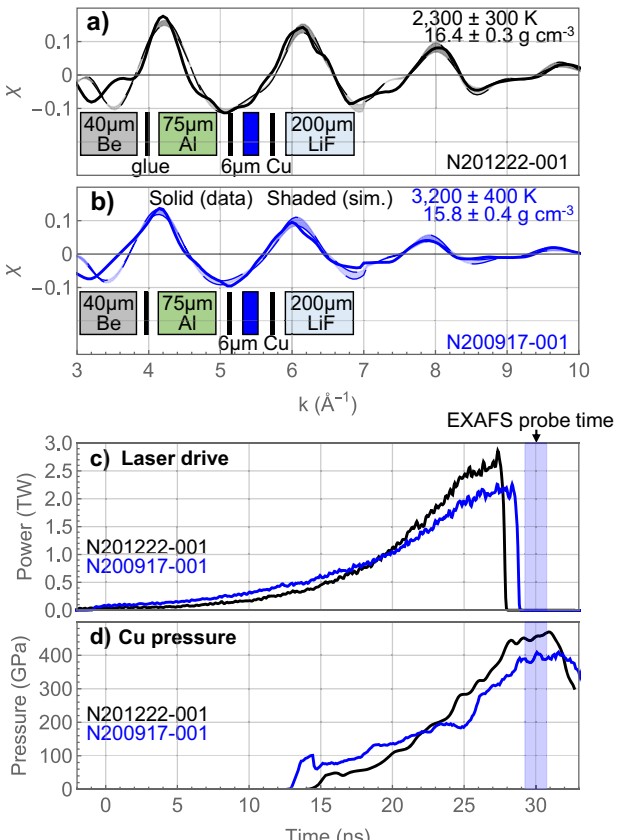

**Fig. 2 | Experiments varying shock strength. a** and **b** Measured Cu EXAFS spectra near 400 GPa in targets with a LiF window. The Cu target package used on each shot is denoted in the diagram under each lineout (vertical black bar represents a glue layer). Data is represented by solid lines. The shaded regions represent the best-match synthetic EXAFS signals (and temperature uncertainty) generated from MD simulation. N201222 (black) in **a** is driven with a lower initial shock into the Cu sample as compared to N200917 (blue) in (**b**), leading to a lower temperature. The data uncertainties in temperature and density are determined from fits to the EXAFS data from a single measurement. Laser power histories are shown in (**c**). Pressure histories (**d**) in the Cu sample were determined from characteristic analysis of VISAR data. Relative timing has been adjusted such that the EXAFS probe time is the same at 30 ns. Source data are provided as a Source Data file.

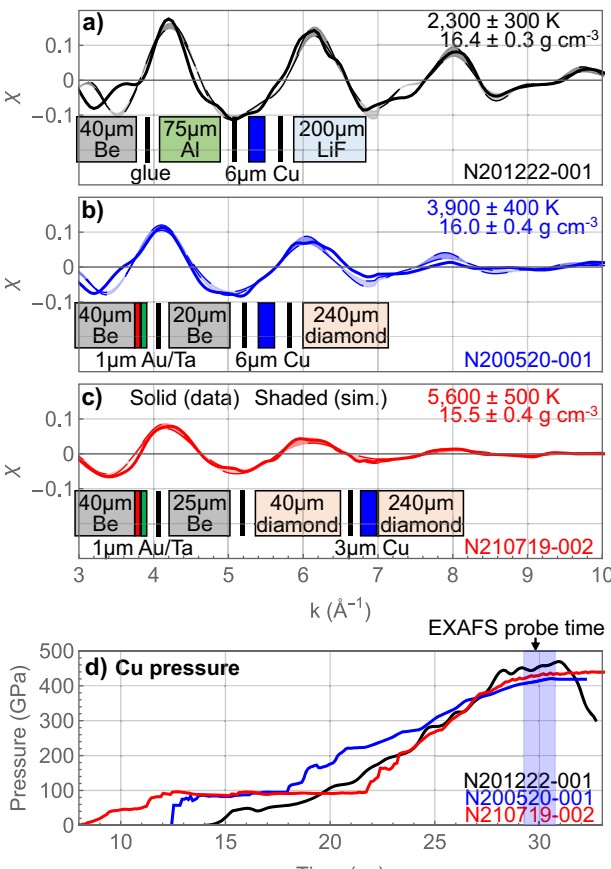

**Fig. 3 | Experiments with different adjacent layers. a**–**c** Measured Cu EXAFS spectra near 400 GPa in targets driven with a ramp drive. The Cu target package used on each shot is denoted in the diagram under each lineout (vertical black bar represents a glue layer). Data is represented by solid lines. The Cu sample in N201222 (black) (**a**) is adjacent to LiF, whereas the Cu sample in N200520 (blue) (**b**) and N210719 (red) (**c**) is adjacent to one or two diamond layers, respectively. The shaded regions represent the best-match synthetic EXAFS signals (and temperature uncertainty) generated from MD simulation. The data uncertainties in temperature and density are determined from fits to the EXAFS data from a single measurement. **d** Pressure histories in the Cu sample determined from characteristic analysis of VISAR data. Relative timing has been adjusted such that the EXAFS probe time is the same at 30 ns. Source data are provided as a Source Data file.

temperature EXAFS measurement. This EXAFS measurement (N210719) is shown in Fig. 4a with synthetic EXAFS spectra calculated from MD simulations of fcc and bcc structures.

We find that the MD-calculated fcc EXAFS signal is a much better match to the measured data near 400 GPa, and has markedly different amplitude and amplitude decay compared to the MD-calculated bcc EXAFS signal. The differences in peak amplitude values and positions are significant given that the experimental uncertainty is 0.5–1.0%. All measured EXAFS spectra in Figs. 2 and 3 are consistent with fcc Cu.

We also measured EXAFS of ramp-compressed Cu near 1 TPa (Fig. 4b). The synthetic EXAFS signal for fcc (9000 K) and bcc (8000 K) at this pressure do not differ sufficiently to identify the structure. The Cu sample in this experiment is designed to be ramp-compressed to ~1 TPa, and this is reflected in the Cu pressure history determined from velocimetry (Fig. 4d). The estimated Cu EXAFS temperature near 1 Pa following a ramp-compression path using MD simulations is 8500 ± 1500 K, significantly higher than the expected temperature along the isentrope. The Cu layer for the data near 1 TPa is sandwiched between two diamond layers. Therefore, this high temperature is consistent with the observation near 400 GPa that the Cu sample is hotter when adjacent to diamond layers. An experiment summary table is provided in Supplementary Table 1.

Figure 5 summarizes the EXAFS results near 400 GPa in the context of the existing literature. The Cu phase diagram shows the principal Hugoniot and isentrope (calculated from an initial 50-GPa shock) from SESAME 3336, along with calculated melt curves[30–33] and predicted fcc–bcc phase boundaries[25,26,34]. For shock-compressed Cu (along the principal Hugoniot)[34,35], a transformation from ambient fcc to bcc structure is experimentally observed near 180 GPa using X-ray diffraction. Along a quasi-isentropic compression path, Cu was experimentally observed to remain in the fcc phase up to 1.15 TPa[13] (these X-ray diffraction measurements did not have accompanying temperature constraints and therefore not included in Fig. 5).

Our Cu EXAFS data near 400 GPa are consistent with the fcc structure remaining stable up to ~5600 K, as compared with theoretical calculations which predicted a fcc–bcc phase transformation in the 5000–6000 K range[25,26]. This set of measurements demonstrates the unique capability of EXAFS (in combination with velocimetry measurements) to constrain the structure, temperature, density, and pressure for the equation of state construction and stable crystal structure determination.

Furthermore, the density, temperature, and pressure determined from simultaneous EXAFS and VISAR measurements provide a

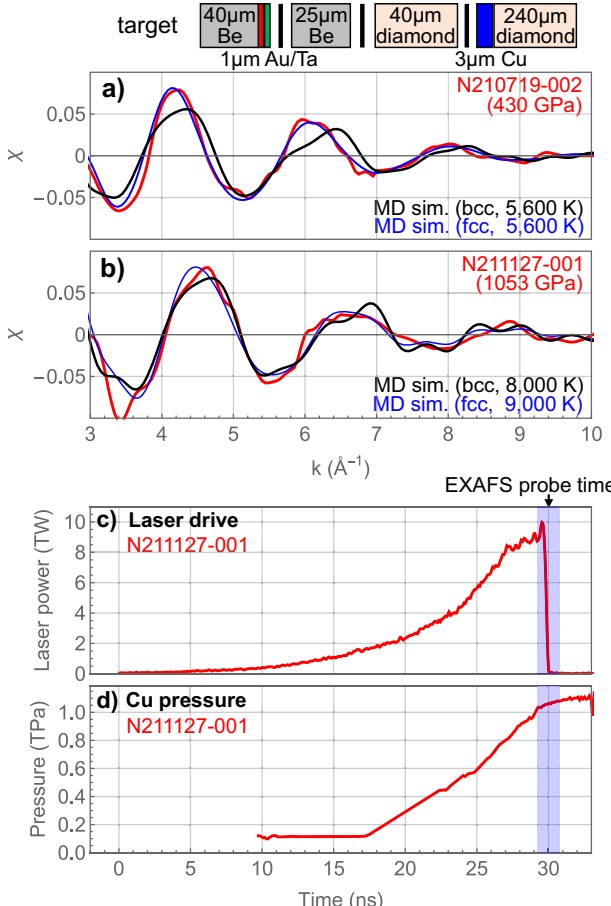

**Fig. 4 | Cu lattice structure.** Comparison of measured EXAFS (red) with MD-calculated EXAFS generated from a fcc (blue) or bcc (black) structure at **a** 430 GPa and **b** 1053 GPa. The laser drive and Cu pressure determined from velocimetry for the experiment near 1 TPa are shown in **c** and **d**, respectively. Both experiments near 400 GPa and 1 TPa used the same target geometry with the Cu sample between two diamond layers (top). Source data are provided as a Source Data file.

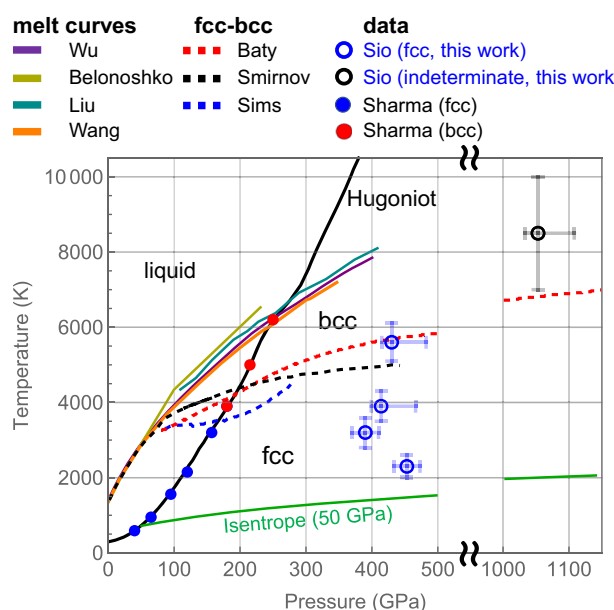

**Fig. 5 | Cu phase space.** Temperature determined from EXAFS, and pressure determined from velocimetry in this work in the context of existing Cu data in the literature. The Hugoniot (black solid) and the isentrope (green solid, calculated from an initial 50 GPa shock) are plotted along with calculated melt curves[30–33], fcc–bcc phase boundaries[25,26,34], and X-ray diffraction data along the Hugoniot[34,35]. The data uncertainties in temperature and pressure are determined from the EXAFS data and VISAR data, respectively, from a single measurement.

complete set of equation-of-state data without assuming particular thermal dynamic paths. We compare the density and temperature as determined from EXAFS with Cu equations of state LEOS 290 and SESAME 3336 in Fig. 6. The equations of state are expressed as isobars with the pressure as determined from VISAR characteristic analysis and the shaded regions representing pressure uncertainty. Because the EXAFS experiments are at slightly different pressures, they are compared with the isobar at their respective pressures. LEOS 290 is found to be in better agreement with the EXAFS measurements as compared to SESAME 3336.

## Discussion

We now discuss two unexpected experimental observations from the three EXAFS experiments near 400 GPa in Fig. 3: (1) Cu temperature is higher when using a diamond window as compared to a LiF window, and (2) Cu temperature further increases when the Cu sample is adjacent to two diamond layers versus a single diamond layer. To explain this surprising temperature behavior in Cu, we examine the expected contributions from X-ray source heating, shock heating, Cu strength, ablation plasma heating, and thermal conduction from diamond.

The first three potential sources can be excluded using existing data and calculations. The Cu sample heating from the Ti X-ray source is experimentally determined to be ~100 K for the target package used in N200520 (Fig. 3, through EXAFS measurements of an undriven

target. These EXAFS data are compared against synchrotron data and fitted to constrain the heating contribution from the Ti X-ray emission (see Supplementary Fig. 4). Shock heating is another potential mechanism, although one that is taken into account in radiation-hydrodynamic HYDRA simulations and experimentally constrained by VISAR data. All the experiments near 400 GPa have an initial ~40–85 GPa shock in the Cu sample, which is confirmed by VISAR data. Cu strength can potentially contribute to Cu heating from plastic work, but this would not explain the observed temperature sensitivity to different adjacent layers.

Heating from ablation plasma X-ray is a potential concern, as the expected X-ray transmission through the target package is ~2× higher in the diamond targets (N200520 and N210719) than in the LiF targets (N200917 and N201222), depending on the ablation plasma emission spectrum. This difference in X-ray transmission through the target package and X-ray absorption in the Cu sample is due to differences in the X-ray shields (1 μm Au/Ta for the diamond package, and 75 μm Al for the LiF package). The radiation heating contribution to the Cu temperature is estimated to be ~200 K in 2-D hydrodynamic simulations. Without invoking unreasonably large simulation errors, heating from ablation plasma emission cannot explain the observed 1600 K Cu temperature increase from LiF (N2012222) to diamond (N200520). Further, ablation plasma heating cannot explain the additional heating measured when a diamond layer is added adjacent to the Cu and facing the ablation surface (N210719, as compared to N200520). For this case, as expected, simulations show little effect of the additional diamond (and thinner Cu sample) on sample heating from ablation plasma emission.

This leaves thermal conduction as the remaining plausible explanation. Heat conduction from a hotter diamond layer into the Cu layer can qualitatively explain both unexpected experimental observations. We calculate the plastic work heating in diamond from[11] $\Delta T_{\text{plastic}} = f_{\text{TQ}} V_0 \int \frac{1}{C} Y \frac{d\epsilon^P}{d\eta} d\eta$, where the temperature increase from plastic work heating $\Delta T_{\text{plastic}}$ is related to the Taylor–Quinney factor[36] $f_{\text{TQ}}$, initial volume $V_0$, heat capacity $C$[37], strength[11] $Y$, and the

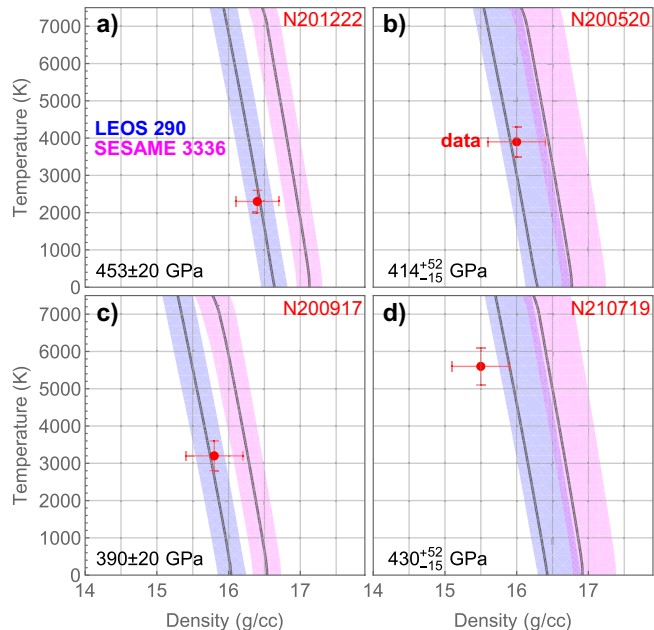

**Fig. 6 | Comparison with Cu equations of state.** Density and temperature determined from EXAFS measurements (red) as compared to Cu equations of state LEOS 290 and SESAME 3336 for experiments **a** N201222-001, **b** N200520-001, **c** N200917-001, and **d** N210719-002. The equations of state are expressed as isobars with the pressure as determined from VISAR characteristic analysis and the blue and pink shaded regions representing pressure uncertainty. The data uncertainties in temperature and density are determined from the EXAFS data and the pressure uncertainty is determined from VISAR data from a single measurement.

plastic strain $\epsilon^p = 2/3\log(\eta) - Y/2G$, with $G$ being the shear modulus[38] and $\eta$ corresponds to the compression ratio $\rho/\rho_0$. The Taylor–Quinney factor $f_{TQ}$ relates the fraction of plastic work converted to heat and is taken to be unity in this work. We start with the diamond window pressure history near the Cu interface inferred from the characteristic analysis of the VISAR data. For each increment of total strain, plastic work and plastic work heating are calculated. The diamond temperature is updated at each iteration and used in the calculations of temperature-dependent quantities (heat capacity, shear modulus, etc.). In this model, the diamond can reach the high temperatures observed in the Cu sample (~5000 to 6000 K). The thinner Cu sample and lack of glue layer on one side in N210719 versus N200520 may also have contributed to faster thermal equilibrium with adjacent layers. For Cu and diamond temperatures to equilibrate over a few nanoseconds requires thermal conductivity both in the Cu sample and between Cu and diamond to be >10 × ambient values, which are plausible but difficult to constrain due to sparse thermal conductivity data in these extreme conditions[39].

Figure 7 compares the temperature determined from EXAFS with 1-D radiation-hydrodynamic HYDRA[40] simulations. The shaded regions represent the spread in pressure and temperature within the Cu sample in the simulations. A commonly used Lee–More thermal conductivity model[41] is employed. The relatively larger spread in temperature in N201222 and N200520 is because the Cu layer is between two different layers at different temperatures. We observed that for the experiment with the LiF window (N201222), HYDRA-simulated temperatures are reasonably close to the isentrope starting from a 50 GPa initial shock. HYDRA simulations with an improved Steinberg-Guinan model[38] for diamond do predict higher diamond temperature as compared to the Cu, but still lower as compared to the inferred Cu EXAFS temperature.

An EXAFS temperature is inferred from thermal disorder in the lattice, but the measurement itself cannot distinguish between thermal disorder (from thermal motions of the atoms) and static disorder (from dislocations, mixed phases, grain boundaries, or other deviations from a perfect lattice) which may lead to an inflated temperature estimate. Experimentally, we note that the temperature inferred from N201222 (Fig. 7) is very close to HYDRA simulation (within 200 K), indicating that static disorder is likely small relative to thermal disorder. A static disorder component may increase the absolute temperatures inferred from EXAFS, but would not affect the observed relative temperature differences between experiments with different adjacent materials.

In many planar dynamic-compression experiments for a wide range of materials[13,21,42–48] including compound, metal, and metal alloy, diamond is a common material in the target either as a pusher or window adjacent to the sample of interest. The results from our experiments can provide important additional information for understanding heat flow in past experiments and for the designs of future experiments using diamonds. For example, our finding suggests that the assumption of minimal thermal transport over a several-ns or longer timescale should be evaluated on a case-by-case basis.

In summary, the temperature, density, pressure, and structure of copper dynamically ramp-compressed are determined using measured EXAFS combined with ultrafast velocimetry and MD simulations. This technique is also expected to be generally applicable for equation-of-state measurements of mid-$Z$ materials important for planetary science and geoscience, and relevant due to its implications for using HED platforms to develop pressure standards and to probe exotic phases that are tests of the most fundamental aspects of materials physics and quantum mechanics. The observed temperature sensitivity to adjacent layers in these multi-layered targets is both unexpected and relevant for future experimental designs, emphasizing the importance of temperature diagnostics in different dynamic-compression platforms, as well as the need for improved understanding of thermal transport and dissipative processes in high-energy-density conditions.

## Methods
### Target preparation and metrology
Several target designs are used in these experiments, which follow the general pattern of a beryllium ablator, a heat shield, and then the Cu sample between a pusher in front, and a window at the rear side. For accurate modeling of the experiment, high-accuracy metrology is required. A double-sided non-contact ZYGO white-light interferometer is used to measure a 3-D thickness map for each component in these multi-layered targets prior to assembly and during assembly after each subsequent layer is added, with average thickness in the region of interest determined with an uncertainty of ±0.2 μm. Average glue thickness (typically ~0.5–1.5 μm) is assessed by subtracting the individual layer thicknesses from the total thickness after gluing, with an uncertainty of ±0.3 μm. Cu and Al films used in the target are procured from Goodfellow[49]. The exceptions are the 3 μm Cu layers used on N210719 and N211127, which are vapor deposited onto the diamond substrate to avoid large thickness variance in very thin foils. The density of the Cu deposited layer is measured to be 99% of the bulk value.

### EXAFS measurements and temperature determination
Up to 88 lasers (351 nm, 4.8 kJ per beam) are used to drive a Ti foil X-ray source[27] on both sides, producing a bright X-ray continuum around the Cu K-edge energy region. The X-ray source is positioned ~25 mm away from the Cu sample, with a 60-μm-thick Zn X-ray heat shield in between. The Zn filter, with a K-edge at 9.66 keV, plays an important role in reducing X-ray flux above the Cu K-edge EXAFS region (9.0–9.5 keV) and minimizing sample heating by the X-ray source.

A high-resolution toroidal crystal spectrometer (HiRAXS)[50] with 3–4 eV resolution between 8.9 and 9.8 keV was designed and fielded to

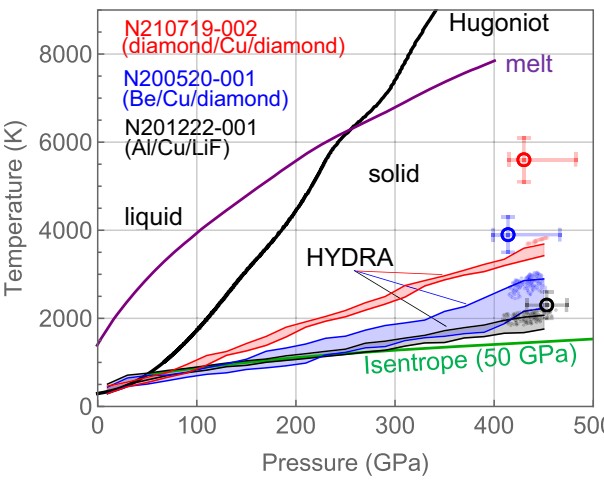

**Fig. 7 | Comparison with hydrodynamic simulations.** The Hugoniot (black) and the isentrope (green, calculated from an initial 50 GPa shock) are plotted along with a calculated melt curve[30] (purple) and data (open circle). Quantities for the three experiments N201222, N200520, and N21071 are represented in black, blue, and red, respectively. The shaded regions represent the spread in pressure and temperature within the Cu sample in the HYDRA simulations. The array of points near 400 GPa represents the spread in pressure and temperature within the Cu sample in the HYDRA simulation over the duration of the EXAFS probe, with the temperature contribution from backlighter heating included. The data uncertainties in temperature and pressure are determined from the EXAFS data and VISAR data, from a single measurement.

measure the X-ray transmission spectrum through the Cu sample. Measured spectrum is smoothed over the instrument resolution, converted to absorption coefficients ($\mu$), background subtracted, and normalized to obtain the EXAFS data $\chi = (\mu - \mu_0)/\mu_0$, where $\mu_0$ is the non-oscillatory component of the absorption coefficients. This is usually plotted as a function of the photoelectron wavenumber $k = \sqrt{(2m_e(E - E_0)}./\hbar$, where $E$ is the photon energy, $E_0$ is the energy of the absorption edge, $m_e$ is the electron mass, and $\hbar$ is the reduced Planck constant. EXAFS data processing details can be found in Supplementary Note 3.

We constrain the sample temperature using molecular dynamics (MD) calculations. Equilibrium MD simulations of a perfect fcc (or bcc) Cu lattice at specific density and temperature conditions are performed using the MD code large-scale atomic/molecular massively parallel simulator (LAMMPS)[51] with the Mishin potential[52]. Synthetic EXAFS signals at different temperatures are calculated from MD simulations of atomic positions using the FEFF[53,54] package and directly fitted to the measured EXAFS data by minimizing the difference squared between measured and synthetic EXAFS. The MD simulations are validated by state-of-the-art DFT-MD simulations, and anharmonicity is self-consistently included in these simulations and synthetic EXAFS spectra. This fitting procedure captures well the overall EXAFS oscillation period and amplitudes; minor differences between measured and synthetic EXAFS are partly due to measurement uncertainty. More information on MD simulations, EXAFS processing, and data uncertainty is presented in Supplementary Note 4.

**VISAR measurements**
Simultaneous velocimetry measurements of the Cu/window interface and/or the window free surface are made alongside the EXAFS measurements using a velocity interferometer system for any reflector (VISAR)[28] system. The NIF dual-channel, line-imaging VISAR system detects Doppler shifts of a 660-nm optical probe reflecting off a moving surface (in these experiments, either the Cu-window interface or the window free surface). An angled mirror (25-μm thick and made

from black nano-diamond polished to optical quality) is positioned behind the target to reflect optical light from the back of the target to the VISAR. This thin diamond window does not interfere with the EXAFS measurement as its X-ray transmission is smooth in the EXAFS region.

The Cu pressure at the time of EXAFS measurement is determined through the measured velocities at the Cu/window interface and/or window free surface using a backward hydrodynamic characteristic analysis[55,56]. More detailed information on VISAR measurement and pressure determination is presented in Supplementary Note 2.

**Reporting summary**
Further information on research design is available in the Nature Portfolio Reporting Summary linked to this article.

## Data availability
The pressure, density, and temperature data determined from EXAFS and velocimetry are provided in Supplementary Table 1. The data generated in this study have been deposited in the Figshare database under accession code https://doi.org/10.6084/m9.figshare.24240034. Source data are provided with this paper.

## Code availability
There are no custom codes or mathematical algorithms central to the conclusions. The publicly available XAFS analysis code DEMETER can be found here: https://bruceravel.github.io/demeter/.

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

## Acknowledgements

This work was performed under the auspices of the U.S. Department of Energy by Lawrence Livermore National Laboratory under Contract DE-AC52-07NA27344. We gratefully acknowledge the LLNL target fabrication and target diagnostics teams, as well as the NIF operation team for the execution of these experiments. LLNL's AnalyzeVISAR code was used to analyze the VISAR data.

## Author contributions

H.S. was the principal investigator for the experimental campaign. and has performed data analysis and interpretation. A.K. and Y.P. were supporting principal investigators for the experimental campaign, and have performed data analysis and interpretation. D.G.B. performed LASNEX simulations and experiment design. R.E.R. performed MD simulations and interpretation. S.A.B. performed DFT-MD simulations and interpretation. M.M. performed a VISAR analysis. D.E.F. performed characteristic analysis of the VISAR data and interpretation. F.C. contributed to the experiment design and VISAR measurements. N.B. and S.V. performed target design.

N.I. performed laser pointing design. A.K., G.K., and T.E.L. performed HYDRA simulations and interpretations. M.B., P.C.E., L.G., K.W.H., N.O., M.B.S., S.S., D.B.T., and B.K. contributed to the characterization of the HiRAXS spectrometer for EXAFS data analysis. Y.P., D.K.B., J.H.E., J.M.M., W.H., K.L.G., and A.M. provided supervision and technical guidance for the project. Y.P., D.K.B., J.H.E., J.M.M., W.H., R.H., O.L.L., C.J.W., H.-S.P., and B.A.R. contributed to the initial experiment conception and design.

## Competing interests

The authors declare no competing interests.
