## [Peer Review File · Nature Communications]

REVIEWER COMMENTS

Reviewer #2 (Remarks to the Author):

This very interesting manuscript reports novel theoretical and experimental results for the thermodynamic properties of dynamically compressed matter in extreme conditions. As detailed in my first report, I believe this manuscript represents research of very high-quality on a topic of broad importance, which is carried out with state-of-the-art theory and experiment.

In their revision the authors have addressed most of the additional points where I felt clarification is desirable.

However, their comment on the near-edge data in the supplementary material should be improved. This point is relevant since the near edge behavior can provide an independent approximate check on the temperature of a system. While this behavior is different for K and L-edges, one expects the K-edge-jump around 8978 eV to broaden systematically with increasing temperature consistent with the Fermi distribution. However, this behavior is not evident in the near-edge data shown in the rebuttal or in Fig. 3 in the supplementary material. Also, the fine structure between 9005 and 9010 eV for the 5600 K plot is larger than that of the lower temperature data. Since these observations lead one to question the near-edge data or its interpretation, further clarification is needed.

Reviewer #2

This very interesting manuscript reports novel theoretical and experimental results for the thermodynamic properties of dynamically compressed matter in extreme conditions. As detailed in my first report, I believe this manuscript represents research of very high-quality on a topic of broad importance, which is carried out with state-of-the-art theory and experiment.

In their revision the authors have addressed most of the additional points where I felt clarification is desirable. However, their comment on the near-edge data in the supplementary material should be improved. This point is relevant since the near edge behavior can provide an independent approximate check on the temperature of a system. While this behavior is different for K and L-edges, one expects the K-edge-jump around 8978 eV to broaden systematically with increasing temperature consistent with the Fermi distribution. However, this behavior is not evident in the near-edge data shown in the rebuttal or in Fig. 3 in the supplementary material. Also, plot is larger than that of the lower temperature data. Since these the fine structure between 9005 and 9010 eV for the 5600 K observations lead one to question the near-edge data or it's interpretation, further clarification is needed.

We very much appreciate this feedback and address the near-edge data in more details below.

We first clarify the impact of measurement resolution on the interpretation of near-edge data measured on NIF. **Figure 1** shows the Cu K-edge data (near-edge) in ambient conditions taken at the National Ignition Facility (NIF, this work) and at a synchrotron facility with $<1\text{eV}$ resolution for comparison. To determine the spectral resolution, the synchrotron XANES spectrum is blurred by a Gaussian function with various Full-Width-Half-Maximum (FWHM) to compare with our undriven data. We found that a FWHM = 4 eV matches the undriven data best (**Fig. 1**). While the spectrometer can achieve a resolution of 3 eV [1], we used a configuration with a larger crystal area to increase signal to noise, which slightly degraded resolution.

Fig. 1: Cu K-edge data (near-edge) in undriven conditions taken at the National Ignition Facility (NIF, this work) and at a synchrotron facility (blurred with a Gaussian function with different FWHM).

As we see in **Fig. 2a**, this measurement resolution has a small but noticeable effect on the near-edge structure, and may blur out small differences in the near-edge from temperature effect. This measurement resolution is already taken into account in analysis of all EXAFS data. Reviewer #2 correctly noted a discrepancy in the Cu K-edge fine structure between 9005 and 9010 eV for the 5,600 K observations in the previous response. We reexamined and carefully renormalized all the data for better comparison, summarized in **Fig. 2a**.

Fig. 2: a) Cu K-edge data (near-edge) of Cu samples compressed to 400 GPa at different temperatures. b) the Fermi-Dirac distribution at different temperatures (dashed), and when blurred over the measurement resolution (solid).

As the 400 GPa NIF data discussed in this work covers a relatively small temperature range (from 2,300 K to 5,600 K), we examine the expected broadening of the Fermi-Dirac distribution with increasing temperature. **Figure 2b** plots the Fermi-Dirac distributions at 2,000 K, 6,000 K, and 10,000 K (all dashed), and also when the distributions are blurred by the measurement resolution (solid). We observe that the difference in the Fermi-Dirac distributions of different temperatures is difficult to distinguish with the NIF measurement resolution (4 eV) included. In experiments [2-4] at higher temperature (up to $\sim 20,000$ K) and higher measurement resolution (~ 1 -2 eV), temperature sensitivity in the near-edge structure is more readily observed.

The above data and discussion are summarized in a new section in the supplementary materials (pg 9-10).

- [1] S. Stoupin *et al.*, *Rev. Sci. Instrum.* **92**, 053102 (2021)
- [2] A. Mančić, *et al.*, *Phys. Rev. Lett.* **104**, 035002 (2010)
- [3] B. I. Cho, *et al.*, *Phys. Rev. Lett.* **106**, 167601 (2011)
- [4] N. Jourdain *et al.*, *Phys. Rev. B* **97**, 075148 (2018)

REVIEWERS' COMMENTS

Reviewer #2 (Remarks to the Author):

As noted in previous reviews, this very interesting manuscript reports novel theoretical and experimental results for the thermodynamic properties of dynamically compressed matter in extreme conditions. As detailed in my first report, this manuscript represents research of very high-quality on a topic of broad importance, which is carried out with state-of-the-art theory and experiment, and a careful analysis.

In their revision the authors have done a careful re-analysis of the near edge data, which is now consistent with expected behavior based on thermal broadening and experimental resolution. The reanalysis is carefully documented in the updated supplementary material. This revision fully addresses my previous concern about the quality of the near edge data the authors' interpretation.

I can now recommend publication in Nature Communications in its present form.